# There You Are! Automated Detection of Indris’ Songs on Features Extracted from Passive Acoustic Recordings

**DOI:** 10.3390/ani13020241

**Published:** 2023-01-09

**Authors:** Davide Ravaglia, Valeria Ferrario, Chiara De Gregorio, Filippo Carugati, Teresa Raimondi, Walter Cristiano, Valeria Torti, Achaz Von Hardenberg, Jonah Ratsimbazafy, Daria Valente, Cristina Giacoma, Marco Gamba

**Affiliations:** 1Department of Life Sciences and Systems Biology, University of Torino, 10123 Torino, Italy; 2Conservation Biology Research Group, Department of Biological Sciences, University of Chester, Chester CH1 4BJ, UK; 3Groupe d’Étude et de Recherche sur les Primates de Madagascar (GERP), Fort Duchesne, Antananarivo 101, Madagascar; 4Parco Natura Viva Garda Zoological Park, 37012 Bussolengo, Italy

**Keywords:** bioacoustics, passive acoustic monitoring, loud calls, automated detection, species recognition, CNN, data augmentation, transfer learning, singing primates, *Indri indri*

## Abstract

**Simple Summary:**

Identifying vocalisations of given species from passive acoustic recordings is a common step in bioacoustics. While manual labelling and identification are widespread, this approach is time-consuming, prone to errors, and unsustainable in the long term, given the vast amount of data collected through passive monitoring. We developed an automated classifier based on a convolutional neural network (CNN) for passive acoustic data collected via an in situ monitoring protocol. In particular, we aimed to detect the vocalisations of the only singing lemur, *Indri indri*. Our network achieved a very high performance (accuracy >90% and recall >80%) in song detection. Our study contributes significantly to the automated wildlife detection research field because it represents a first attempt to combine a CNN and acoustic features based on a third-octave band system for song detection. Moreover, the automated detection provided insights that will improve field data collection and fine-tune conservation practices.

**Abstract:**

The growing concern for the ongoing biodiversity loss drives researchers towards practical and large-scale automated systems to monitor wild animal populations. Primates, with most species threatened by extinction, face substantial risks. We focused on the vocal activity of the indri (*Indri indri*) recorded in Maromizaha Forest (Madagascar) from 2019 to 2021 via passive acoustics, a method increasingly used for monitoring activities in different environments. We first used indris’ songs, loud distinctive vocal sequences, to detect the species’ presence. We processed the raw data (66,443 10-min recordings) and extracted acoustic features based on the third-octave band system. We then analysed the features extracted from three datasets, divided according to sampling year, site, and recorder type, with a convolutional neural network that was able to generalise to recording sites and previously unsampled periods via data augmentation and transfer learning. For the three datasets, our network detected the song presence with high accuracy (>90%) and recall (>80%) values. Once provided the model with the time and day of recording, the high-performance values ensured that the classification process could accurately depict both daily and annual habits of indris‘ singing pattern, critical information to optimise field data collection. Overall, using this easy-to-implement species-specific detection workflow as a preprocessing method allows researchers to reduce the time dedicated to manual classification.

## 1. Introduction

In a constantly changing environment witnessing unprecedented species loss, the need for automated systems for environmental surveillance is driving researchers towards using passive acoustics as one of the most widely employed methods for monitoring wild animals [1]. Passive acoustic monitoring (PAM) enables the collection of information about species’ presence, abundance, density, distribution, population status, and dynamics; moreover, PAM allows for investigating the seasonal and geographical variability of vocal behaviour [2]. Using passive acoustics has undeniable advantages: it represents a noninvasive and relatively cheap technology in comparison to direct data collection, and recorders are easy to deploy and can be left in situ for extended periods [3]. However, passive monitoring generates an impressive amount of data that can be difficult and time-consuming to store, process, and analyse [3] and in this regard is more challenging than other monitoring methods. Although most data are still manually processed, in recent years the use of machine learning for automating or at least facilitating the labelling and analysis of bioacoustic data is becoming increasingly relevant [4]. For instance, the combination of bioacoustic monitoring with machine learning algorithms has been proven to provide reliable insights into species distribution, behaviour, habitat preference, and baseline data to quantify advancing modifications related to habitat disruption or climate change [5]. These techniques can be used to better understand target species and, consequently, to plan conservation strategies based on informed decision-making processes [5]. 

In particular, the use of convolutional neural networks (CNNs) for automatically classifying environmental and animal sounds is increasing [6]. For instance, CNNs have been successfully trained to classify the presence and absence of different kinds of sounds, taxonomic groups, and environments, using owl (*Strix aluco* and *Athene noctua*) songs, echolocation calls of different genera of European bats [7], and sperm whale (*Physeter macrocephalus*) echolocation clicks [7,8]. Besides a very high accuracy in spectrogram classification (99.5%), the study by Bermant and colleagues also revealed the potential of these architectures in extracting fine-scale details. Indeed, the network has been trained to classify coda types recorded in different zones (Dominica and Eastern Tropical Pacific, with 23 and 43 coda types, and the coda type classification achieved accuracy values of 97.5% and 93.6, respectively [8]). Moreover, in a further task of vocal clan discrimination, the network achieved an accuracy of 95.3% for two Dominican clans and 93.1% for four Eastern Tropical Pacific clans [8]. Automatic detection and classification using a CNN on spectrograms have also been applied to bearded seal (*Erignathus barbatus*) vocalisations in the north-eastern Chukchi Sea [2]. The results indicated an accuracy above 95.5% in classifying between noise and one vocal type; when generalising the network performance to data recorded at different locations, the authors found a precision above 89.2% [2]. Another study surveying bat (several species belonging to the genera *Myotis, Plecotus, Pipistrellus, Nyctalus, Barbastella, Eptesicus,* and *Vespertilio*) activity relied on the extraction of Mel-frequency cepstral coefficients (MFCC) that were converted into images and then fed to a CNN to classify bat calls and noise. Here, the network reached accuracy values of 99.7% in the binary classification of noise and bat calls and 96% for the sole species classification [9].

When employing these algorithms, however, the effectiveness of the best practices (e.g., data type, data processing, and data augmentation) is still debated [4]. For instance, in a study employing a CNN to classify calls of the yellow-vented warbler (*Phylloscopus cantator*) and the rufous-throated wren-babbler (*Spelaeornis caudatus*) [5], the authors successfully performed data augmentation for calls of both species, increasing the training set size and boosting model performance. The network provided indications of species activity and abundance in multiple ecosystems [5]. Data augmentation for improving the accuracy of automated audio classification has also been used for bird (many species -retrieved from XenoCanto [10]) and cat (*Felis catus*) sounds [11,12]. The authors processed the data by applying data augmentation to both raw signals and their spectrograms, finding a maximisation in the classification performance. However, the authors also warned that using augmentation methods developed for images might be useless or detrimental [12]. Machine learning algorithms are also gaining support for analysing primate calls. For instance, a recent study employed an end-to-end feedforward CNN to detect and classify calls of captive common marmosets (*Callithrix jacchus*) using noisy spectrograms without performing any preprocessing of the data [13]. The network was not only able to detect calls and noise but also to classify call types and identify the callers [13]. Interestingly, this study applied data augmentation methods (random noise and amplitude modification) as well, but the authors found that those did not improve the network performance [13]. Another work relied on convolutional recurrent neural networks for the acoustic detection of both the buttress drumming and vocalisations used in the long-distance communication of chimpanzees (*Pan troglodytes verus*), namely pant-hoots and screams [1]. Here, the authors designed a pipeline aiming to compensate for class imbalance and found that spectrogram denoising and standard binary cross-entropy were effective in improving performance for both target classes. Conversely, the manipulation of class imbalance via resampling either worsened or did not affect the network performance [1].

Besides being employed on long-distance calls, these methods can also be used to classify vocal sequences of the so-called singing primate species, a small group of primates that are able to emit complex long-distance vocalisations that are termed songs [14,15]. In fact, neural networks have been proven to be excellent in automatically classifying passively recorded calls of Hainan gibbons (*Nomascus hainanus*) through a CNN trained to classify Mel-frequency scale spectrogram images [4]. To address data scarcity issues related to CNNs, which often occur while studying rare species, transfer learning, a technique using a model trained on a given dataset to predict a second one, has been proposed. A recent study focused on targeting the calls of two critically endangered primate species, the Hainan gibbon (*Nomascus hainanus*) and the black-and-white ruffed lemur (*Varecia variegata*) [16]. It found that transfer learning starting from an all-purpose image database resulted in an improvement in the classification scores, a network that was easier to design, and a reduction in the manual annotation time for both species [16].

Within primates, lemurs offer a relatively straightforward target for acoustic analysis since they often present typical vocal behaviour [17]. Most extant lemur species (103 of 107) are threatened with extinction, with one-third of the species classified as critically endangered [18]; in such a dreadful scenario, environmental monitoring relying on passive acoustics can ease the collection of regular data on species in their natural habitats [19]. Despite providing precious information, data processing and analysis are extremely time-consuming, especially when the applied procedures are manual. Therefore, automation would be highly valuable [13,19]. Furthermore, the vocal behaviour of the species may represent an additional challenge: vocal activity may vary throughout the time of the day or year (in relation to temperature and mating season, for instance) and because some vocalisations, such as those of singing primates, are often performed in a chorus, which renders the estimation of the number of participants challenging [20].

We aimed to apply machine learning techniques combined with passive acoustics by using the only singing lemur species, *Indri indri,* as a model [15]. Indri is a critically endangered lemur [18] and one of the world’s 25 most endangered primates [21]. The population trend of the species is decreasing, mainly due to hunting and habitat destruction [18]. This species exclusively inhabits the rainforests of Eastern Madagascar, an extremely fragile ecosystem. The country has lost 37% of its natural forest cover over 40 years (1973–2014 [22]), and the projected scenarios, with estimated reductions of up to 93% by 2070, are alarming [22]. Moreover, hunting is becoming an increasing threat to the species’ survival [23]. The dense forest environment where indris live is difficult to access. Therefore, passive methods that complement and improve direct observation and monitoring methods are necessary. For PAM to be efficient, the target species should be easily identifiable and recognizable [3]. Hence, despite indris possessing a rich vocal repertoire [24,25] we focused on their songs as target sounds. Songs are indeed peculiar and powerful species-specific vocal displays [26] that are used in intergroup long-distance communication [27] and travel up to two km from the source of emission [28,29,30]. These features make the species perfectly suitable for an effective investigation relying on PAM [3].

We first aimed to develop an easy-to-train, versatile, and fast classifier that was able to target indris’ songs while identifying the best practices for setting up an efficient tool, including which kind of data to feed to the network (i.e., waveform versus spectrogram images), the amount of data to be manually labelled, and the effectiveness of employing data augmentation and transfer learning [4,16]. Our second aim was to use the developed tool to explore diel and seasonal variation in the calling pattern of the species. Indeed, although indris are known to sing more during the breeding season [31] and certain parts of the day [17], an accurate quantification of the seasonal and diel trends of their singing activity was missing. Passive sensors can be used to describe vocal activity over time spans ranging from hours to seasons [32], and passive acoustics has already been successfully employed for the description of the circadian and seasonal trends of primate species’ vocal activity. For instance, PAM has been used to gain ecological insights and to identify the periods with the highest calling patterns at the diel and seasonal levels in the black and gold howler monkey (*Alouatta caraya* [33]) and the Guyanan red howler monkey (*Alouatta macconnelli* [34]). Moreover, providing precise information on the vocal behaviour of the species is fundamental for designing appropriate monitoring programs. This can, in fact, ease data collection and reduce the sampling effort and therefore the amount of unnecessary data to store and process [32].

## 2. Materials and Methods

### 2.1. Data Collection

This work focused on the Maromizaha New Protected Area (18°56′49″ S, 48°27′33″ E), a rainforest with a tropical climate, including a cool dry season from May to September and a warm rainy season from October to April [35]. Maromizaha spans 2150 hectares at an altitude slightly above 1000 m above sea level and is located within the Ankeniheny-Zahamena Corridor, a large area of protected forest biome in eastern Madagascar. We recorded data from 2019 to 2021 using two types of autonomous recorders. In particular, we deployed ten AudioMoth recorders (LabMaker, Berlin, Germany), a low-cost, small-sized, and low-energy directional acoustic detector [36], and two Song Meter SM4 recorders (Wildlife Acoustics Inc., Maynard, MA, USA) [37]. In 2019 we obtained 28,314 recordings by deploying AudioMoth recorders across 12 sites within the Maromizaha forest (hereafter referred to as ‘AM 2019’). In 2020 and 2021 we collected 12,661 recordings via AudioMoth recorders (hereafter ‘AM 2020/2021’) and 25,468 recordings via Song Meter SM4 recorders (hereafter ‘SM4 2020/2021’), placed across six and four sites within Maromizaha forest, respectively (Appendix A). We set a sampling schedule of 10 min every 30 min from 00:00 to 23:30, yielding 48 daily recordings. As recently shown, this schedule reflects a balance between the need to ensure an accurate acoustic survey and providing a tractable amount of data to process [38]. Across all localities, we totalled 664,430 min of recording. The recordings were registered as 600-s-long files with a sample rate of 48 kHz.

### 2.2. Data Processing

By using Praat 6.0.26 [39], we converted each audio recording into a spectrogram, setting the frequency range from 0 to 2500 Hz. This frequency band has previously been used to study the indris’ song [40] since it contains its fundamental frequency (f0) and first harmonics. At the same time, this frequency band minimizes the chosen range, mitigating the presence of other sounds that could be mistaken as a song. 

We visually inspected individual spectrograms for the presence of the song: a single observer (DR) undertook the classification process for the AM 2019 dataset. Then, four researchers with field experience related to indris’ activity verified this classification. To test for interobserver reliability, two primary observers (DR and VF) evaluated 250 randomly selected files from the AM 2019 dataset, achieving a Cohen’s kappa value of 0.9160 [41]. We computed Cohen’s kappa via Python’s *sklearn.metrics.cohen_kappa_score* module (Python 2009; version 3; CreateSpace, Scotts Valley, CA, USA) [42]. Lastly, a different single observer with field experience (VF) undertook the classification process for the AM 2020/2021 and SM4 2020/2021 datasets. 

We manually labelled all recordings, distinguishing between files with or without song presence, considering all cases where at least a descending phrase (the core part of indris’ songs containing a series of phrases, usually beginning with a note followed by one to four with gradually lower frequencies [43]) was visible as ‘containing songs’. We chose this criterion since the phrase constitutes a shared element of indris’ songs, regardless of the context of emission [44]. We labelled 13.7% of the AM 2019 dataset, 9.3% of the AM 2020/2021 dataset, and the 9.7% of the SM4 2020/2021 dataset as files containing songs. We extracted the energy of each central frequency in a one-third-octave band spectrum to quantify the acoustic features of a particular forest environment [45,46]. Using a decomposition of the spectrum into one-third-octave bands has already proved helpful in studying animal vocalisations [47], including primates [46]. We carried out the extraction of the energy coefficients at 20-s intervals. Therefore, for example, for a 600-s file, we computed 30 spectra and obtained 30 sets of values. This time window allowed for reducing the data while ensuring that most songs, whose average duration is 88.61 + 39.51 s [48], were included more than once in the time intervals. We divided the frequency bands according to the one-third-octave system and kept 18 bands from the 26th to the 43rd. The latter choice was based on indris’ hearing range, which, although not precisely known, likely presents an upper bound between 20 kHz and 30 kHz [49].

Through this process, each original recording was transformed into a 30 × 18 matrix of features (30-time intervals and 18 frequency bands). Noticeably, this matrix was more than 10,000 times lighter than the original waveform (~4 kB vs. ~50 MB) and 1000 times lighter than the exported spectrograms (~4 kB vs. ~4 MB). We then normalised the features extracted from the different bands in a 0-1 interval. In this process, the data occurring in the tails of the distribution of these values were removed and later assigned the values of 0 and 1, respectively. We identified the tails of the distribution as the values outside a specified threshold (either lower or higher), which we determined using the quantiles of the SPL distribution. To determine where to truncate the distribution’s tails, we employed different performance metrics to evaluate the model: *accuracy* (the fractions of positives and negatives classified correctly), *precision* (the fraction of true positives divided by the sum of true positives and false positives), *recall* (sensitivity: the fraction of positive cases classified correctly), and the *F1 Score* (the harmonic mean between precision and recall). We employed these metrics for the forthcoming model evaluations as well.

Facing a class-imbalance problem while using binary cross-entropy loss, we decided to apply data augmentation. Considering each normalised matrix as an input image of a CNN, we sliced each 30 × 18 image into six 5 × 18 portions (Figure 1a) and randomly rearranged them (Figure 1b) to obtain a novel 30 × 18 image (Figure 1c). We opted for 5 × 18 slicing because each sequence corresponds to a slice of 100 s. The resulting slice is therefore wide enough to contain a song or a portion of one, which, in turn, should allow it to be detected as a song. We repeated this process for all files containing songs present in the training set until half of the training set consisted of positive cases. We applied this process to each training set. 

### 2.3. Data Analysis

To build the CNN, we employed a starting architecture consisting of alternating convolutional and max pool layers with two final fully connected layers before the binary output [50]. We then modified the starting architecture and fine-tuned the hyperparameters using 70% of the AM 2019 dataset as a training set and the remaining 20% and 10% as a validation and test set, respectively. We chose to develop the CNN using the Keras library available with TensorFlow [51].

The first change in the architecture to significantly improve the performance metrics was the exclusive usage of convolutional layers, a commonly employed method for small image recognition [52]. We then mapped the hour of the day and the week of the year on which the recording took place on two unit circles. After that, we computed the sine and cosine values obtained on the unit circles; by doing so, we represented every hour and week with a couple of unique numbers that maintained a periodic continuous pattern. Next, the four obtained parameters were provided to the CNN in the fully connected layers and improved the song detection. The last step was fine-tuning the following hyperparameters: the learning rate, the dropout rate, the number of neurons in the fully connected layers, the number of filters, and the filters’ aspect ratio, obtaining a total of 84,357 trainable parameters.

Once we defined the final architecture, we trained the CNN on the AM 2019 dataset, saving the weights obtained for the different layers. Next, we tested the effectiveness of this model using the AM 2020/21 and SM4 2020/21 datasets as test sets. We then enforced transfer learning on these datasets, using, in both cases, 5% and 15% of the datasets as training and validation sets, respectively. The validation set has proven to be crucial to determine the layers to freeze the previously obtained weights and the learning rate value. Since the test set for the AM 2019 dataset was a slim selection of the whole dataset, while for both the AM 2020/21 and SM4 2020/21 datasets it was the training sets that were confined selections, we chose to perform 10-fold cross-validation on the three sets of data to evaluate the averages and fluctuations of the performance metrics.

We then verified the normality of the measured metrics through a Shapiro–Wilk test employed through the Shapiro function in the SciPy package [53]. Based on the obtained results, we performed a paired *t*-test or a Wilcoxon signed-rank test (for normally or non-normally distributed data, respectively) to evaluate whether the metric’s value for each dataset significantly changed before and after data augmentation and transfer learning. The two tests were conducted, respectively, with the *Ttest_rel* and the *Wilcoxon* functions present in the SciPy package [53]. We compared the metric values of AM 2019 before and after data augmentation. Instead, for both AM and SM 2020/21, we exerted the same procedure before and after transfer learning, with or without data augmentation. Lastly, we employed the network to determine the calling pattern of the species in terms of both diel and seasonal activity, deriving their respective probability distributions. Regarding the diel distribution, we used 48 30-min bins, thus equalling the time interval between the starts of the two recordings. Although indri activity is tied to the sunrise [28], we decided to show the daily trend accumulated for the entire year instead of dividing it by the different seasons; at this latitude, in fact, the sunrise time range remains roughly constant [54]. Regarding the seasonal distribution, we converted the date associated with each recording to the respective week of the year, binning the year into 52 weeks. If a week contained less than 250 recordings, we considered it “unsampled” and therefore kept its recordings from the computation of the probability of the sampled weeks. With this process, we obtained both the diel and the seasonal distributions from the ten test sets of the AM 2020/2021 and SM4 2020/2021 datasets obtained from cross-validation. Then, separately for the two datasets, we used the ten values obtained for each bin to compute their respective means and standard deviations. Finally, we determined the expected diel and seasonal distributions for the two different datasets via the manual classification performed on the overall dataset.

## 3. Results

### 3.1. Model Performance

The network we built was able not only to reliably detect indris’ songs but also to do so consistently across the test sets and the considered recording locations, as corroborated by the recall (Figure 2 and Table 1) and the other performance metrics (Table 1). When considering the recording sites, the lowest recall value was related to the location at the outermost boundary of the forest (Figure 2). 

When comparing the metrics before and after data processing, we found differences across the three datasets (see Table 1). In particular, considering the AM 2019 test set, we found an improvement in the recall value after data augmentation (z = 2.80, *p* < 0.01), while all the other metrics showed significant decreases (accuracy: t = −9.70, *p* < 0.01, df = 9; precision: t = −10.75, *p* < 0.01, df = 9; F1 Score: t = −12.17, *p* < 0.01, df = 9). 

For AM and SM 2020/21, we compared the metric values before and after transfer learning with or without data augmentation. The comparison of the AM2020/21 test set showed similar trends when performing either transfer learning or a combination of data augmentation and transfer learning. Specifically, we found that all metrics but recall (after transfer learning: t = −18.85, *p* < 0.01, df = 9; after transfer learning and data augmentation: t = −4.19; *p* < 0.01, df = 9) improved both after transfer learning (accuracy: z = 2.80, *p* < 0.01; precision: t = 27.37, *p* < 0.01; F1 Score: t = 49.78, *p* < 0.01, df = 9) and after combining transfer learning and data augmentation (accuracy: t = 12.26, *p* < 0.01, df = 9; precision: t = 9.69, *p* < 0.01, df = 9; F1 Score: t = 11.78, *p* < 0.01, df = 9).

However, the comparison of the SM4 2020/21 test set showed some differences in the values of the metrics when performing either transfer learning or a combination of data augmentation and transfer learning. In particular, after transfer learning, we found improvements in accuracy (t = 44.75, *p* < 0.01, df = 9), precision (z = 2.80, *p* < 0.01), and the F1 score (t = 11.16, *p* < 0.01, df = 9) but a decrease in the recall value (z = −2.80, *p* < 0.01). On the other hand, when performing both data augmentation and transfer learning, we found an improvement only in the recall value (t = 12.44, *p* < 0.01; df = 9), while accuracy (t = −3.73, *p* < 0.01, df = 9) and precision (t = −4.67, *p* < 0.01, df = 9) decreased significantly. The F1 Score showed a comparable trend (t = −1.89, *p* = 0.09, df = 9).

### 3.2. Indri indri Calling Pattern

When using the network to investigate the calling pattern of the species, we found the highest probability of recording songs during the warm rainy season and the morning hours. The two sets of data (AM 2020/2021 and SM 2020/2021) highlighted congruent results. In particular, we found that considering the SM4 (Figure 3a) or AM (Figure 3c) recordings, the calling behaviour of the species changed throughout the year. Both the manual annotation and the network indicated the lowest probability of recording songs during the cool dry season (May through September) and the highest probability of recording songs during the warm rainy season (October through April), with a steady rise between the 35th and the 52nd week. When considering the daily pattern of singing behaviour, we found slight differences between the two sets of data, with SM4 indicating an increase in the calling activity promptly following the astronomical sunrise, with a peak around 10 am (Figure 3b). On the other hand, AM depicted a different scenario, with nocturnal calling activity and a peak almost corresponding to the astronomical sunrise (Figure 3d).

## 4. Discussion

### 4.1. Model Performance

Our study led to defining an automated tool, the CNN, capable of reliably performing a first data skimming over different datasets. We implemented the detection on a limited group of acoustic features based on the third-octave bands’ spectrum analysis, which proved to be a solid representation of the raw acoustic data. Furthermore, since we conceived the CNN as a preprocessing tool, we evaluated the effectiveness of some best practices, such as data augmentation, transfer learning, and using acoustic features instead of spectrograms, prioritising recall when confronting the trade-off between precision and recall. Essentially, in a successive manual classification of the positively classified cases, we encountered a higher number of false positive occurrences. While we easily discarded them in a second manual classification, we concurrently narrowed the false negatives, minimising the number of overlooked recordings containing songs.

Overall, our results indicate a slightly lower performance compared to similar studies. Indeed, recent work relying on CNN to detect and classify different kinds of vocalisations achieved excellent results. For instance, sperm whale echolocation clicks [8] and bat [9] and bearded seal [2] vocalisations were classified with accuracy values of 99.5%, 99.7%, and 95.5%, respectively. However, these studies relied on the classification of spectrograms [2,8] or images reconstructed from Mel-frequency cepstral coefficients [9]. In contrast, we built a network relying on matrices of features up to 10,000 times lighter than the original waveform. While this ensured the acceleration of the entire detection process and paved the way for more frequent data exchange between research centres, it can, in turn, cause a reduction in the retained information and therefore impact the algorithm’s capacity for recognition. Past findings on data preprocessing showed mixed evidence. In particular, the use of data augmentation methods via random noise and amplitude modification did not improve the performance of a network built to detect and classify calls of captive common marmosets [13]. A comparable pattern emerged from an analysis focused on detecting buttress drumming, pant-hoots, and screams of chimpanzees [1]; the authors found that spectrogram denoising and standard binary cross-entropy helped improve performance, compensating for class imbalance. However, they also found that treating class imbalance through resampling either worsened the network performance or did not affect it [1]. In contrast, data augmentation to treat class imbalance on both raw signals and spectrograms led to maximisation in the classification of birds and cat calls [11,12]. In line with these last works, using the recall values as an index of the network’s performance, we found that data augmentation improved the performance of our network; the recall improved noticeably for two of the test sets, while it slightly decreased for the third one. The opposite trend shown by the accuracy (decreasing after data augmentation for two test sets) is a consequence of the class imbalance mitigated by data augmentation. Since negative cases represented the majority of the sample, the model without data augmentation was better at identifying negative cases, with a consequent increase in the accuracy value. Furthermore, in agreement with the findings of Dufourq and colleagues, who employed transfer learning to address data scarcity issues [16] and data augmentation to improve the detection of the long-distance calls of a threatened species [4], we found that a combination of transfer learning and data augmentation indeed resulted in an improvement in the recall values. Noticeably, when applying these methods to novel data (i.e., submitting data collected from different recorders than those used for the training set to the CNN), the recall significantly improved, aligning with the other measured values of recall. Just as we prioritised the recall values to obtain a pipeline to classify instances later verified through manual classification, different studies might favour precision, accuracy, or other metrics, depending on the tool’s aim. Indeed, traditional performance metrics are valuable but not an absolute index of success [4]. Whatever the case, we suggest regularly tracking the value of the F1 Score, which expresses an equilibrium between precision and recall and can help highlight some configurations over others when so many metrics and possibilities are involved.

### 4.2. Indri indri Calling Pattern

Previous studies combining bioacoustic monitoring with machine learning algorithms provided reliable information regarding species presence, distribution, behaviour, and habitat preference [2,5]. In agreement with this evidence, we found that our network was not only able to detect indri songs with a high degree of accuracy but also to consistently predict both the seasonal and diel variations in the calling pattern of the species. When using the network to investigate the seasonal variation, we found the highest probability of recording songs during the warm rainy season (October through April), with a progressive increase between the 35th and the 52nd week. This was consistent with the literature on indri vocal behaviour, reporting December and January to be the months with the highest general activity, including breeding behaviour, and characterised by more intense singing [28,31]. When considering the singing behaviour daily pattern, the SM4 set indicated an increase in the calling activity immediately following the astronomical sunrise, with a peak around 10 am. On the other hand, the AM set indicated nocturnal singing activity and a peak around 5 am. Although we mainly carried out AM recordings from February to July, the SM4 recordings were carried out year-round and should be considered more accurate. Nonetheless, both results are consistent with previous studies in the wild. Indeed, indri songs were reported not to be randomly scattered throughout the day but usually concentrated early in the morning, with a peak between 8.00 and 9.30 am [17,28]. Furthermore, nocturnal singing is not infrequent during the mating season [28]. 

From a future perspective, narrowing down where a song is given could allow a finely tuned recognition of a particular group’s position and identity. We were working with a limited array of sensors, but denser arrays can indeed inform about the precise placement of a group of singers [55]. Instead, investigating the temporal patterning of songs can represent an indicator of the effects of perturbations over an indri population. We know that anthropogenic noise can affect the temporal occurrence of birdsong [56,57], and singing patterns can provide researchers with essential information on the impacts of the human alteration of natural habitats [58], a problem dramatically present in the current situation in Madagascar.

For instance, although we did not directly test potential differences in song detection among the recording locations, we found differences in the recall values among them, with the lowest values of both recall and song occurrence concerning the device placed at the outermost boundary of the forest. This finding entails two considerations. First, the result could indicate a variation in the detection capacity of the autonomous device associated with a specific location [4], but it primarily provides insight into the behavioural ecology of the species. Second, indris inhabiting the mid-elevation rainforest corridor of Ankeniheny-Zahamena, where Maromizaha Forest is located, are indeed known to be edge-intolerant [18]; the lowest song occurrence could indicate a lower abundance of indris in more degraded patches. In turn, this highlights the applicability of population abundance estimation via acoustic capture–recapture relying on PAM [4]. Other future studies could focus on building tools to discriminate between groups [59] or individuals, given the sexual dimorphic structure [48,60] and the ontogenetic variation of indris’ songs [61], and investigate the impacts of environmental factors known to affect singing behaviour (for instance, rainfall [62]).

## 5. Conclusions

Long-term monitoring generates an enormous amount of data: the manual labelling of hundreds of hours of recordings is virtually unfeasible. In line with recent work, our findings show that the combination of passive acoustics with an automated classifier relying on transfer learning, which requires very little training [4], could be an effective monitoring tool for detecting singing species while saving time, energy, storage space, and both economic and human resources [4]. Furthermore, this approach can be extended to other species whose vocal behaviour is suitable for passive monitoring (i.e., relying on loud calls [3]). Although our results are probably not sufficient to fully automate the detection of indris’ songs, our network does allow us to decrease the time dedicated to manual classification. Furthermore, the pipeline proved to be versatile enough to obtain similar or even better performance after the transfer learning when different sites and recorders were employed, requiring us to manually label only a tiny fraction of the original dataset. Future developments will consider the effectiveness of carrying out the first detection and later retraining the model on the cases assigned with high confidence to both classes and the feasibility of applying transfer learning to loud calls of different species.

In conclusion, our study provides further evidence supporting the efficacy of combining passive acoustics with automated classification algorithms to successfully monitor both the presence and the activity patterns of conservation-priority loud-call species.

## Figures and Tables

**Figure 1 animals-13-00241-f001:**
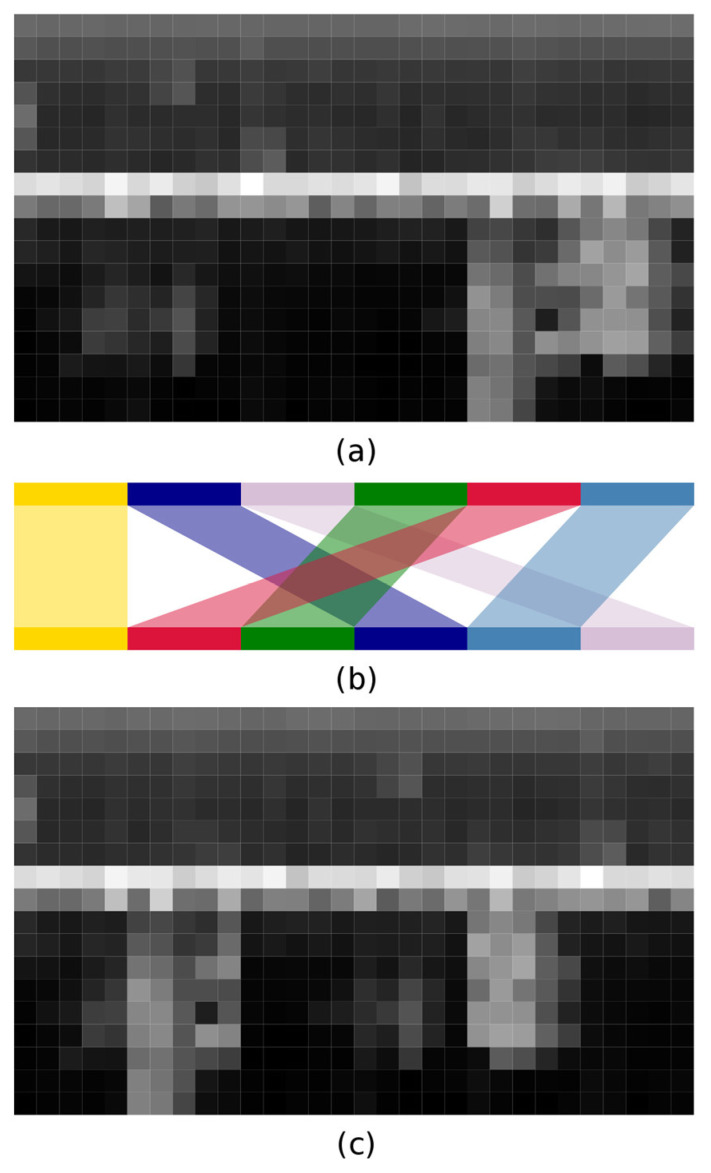
We performed the process used for data augmentation by slicing and rearranging. (**a**) The initial features extracted from the audio file were arranged in a matrix. (**b**) This matrix was sliced into six pieces on the time axis. (**c**) These slices (highlighted with different colours) were then randomly rearranged to create novel input data.

**Figure 2 animals-13-00241-f002:**
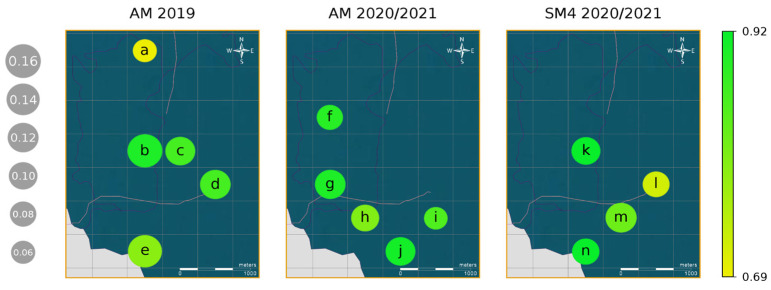
Average occurrence of songs and average recall obtained from 10 iterations of cross-validation, shown for the different recording locations of the three datasets: AM 2019, AM 2020/2021, and SM4 2020/2021. The colour gradient indicates the recall (correctly classified positive cases divided by the total number of positive cases), while the circles’ sizes represent the occurrence (positive cases divided by the total number of the cases). The recording locations shown are those with averages of more than 250 recordings (≈40 h) present in the test set; their exact coordinates and their values of recall and song occurrence are shown in Appendix A.

**Figure 3 animals-13-00241-f003:**
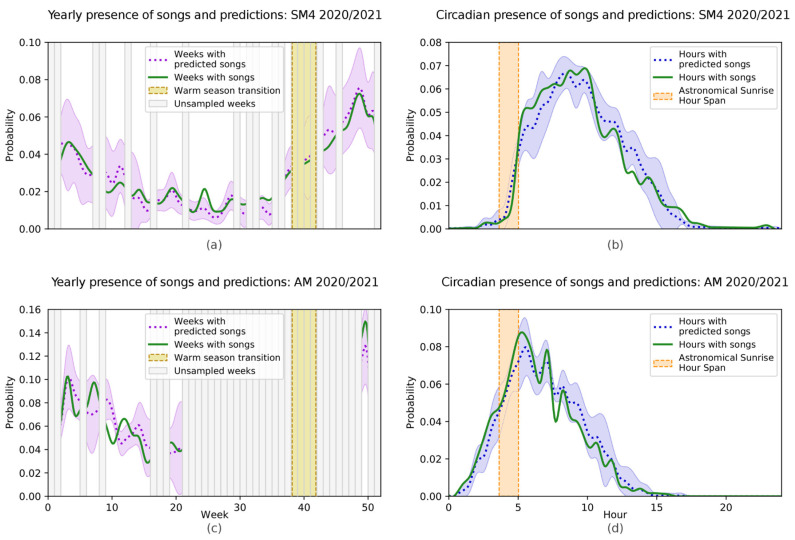
Yearly and circadian trends of the presence of songs and song predictions (positive cases of classification). (**a**,**b**) report data for the SM4 2020/2021 dataset; (**c**,**d**) report data for the AM 2020/2021 dataset. We considered a week unsampled if less than 250 recordings (≈ 40 h) were present; we did not use recordings of unsampled weeks to compute the probabilities for plots (**a**,**c**), related to yearly trends, while we employed all recordings for plots (**b**,**d**), related to circadian trends. In the plots representing the yearly trend, we highlighted the typical period that presents a transition between the cold and warm seasons, while for those concerning the circadian trends we highlighted the astronomical sunrise hour. Dotted lines and shaded areas represent, respectively, the mean and the area between three standard deviations obtained from the ten iterations of cross-validation. We report the trends regarding erroneous and missed predictions in Appendix A, respectively.

**Table 1 animals-13-00241-t001:** Accuracy, precision, recall, and F1 score values were obtained for the test sets of the three sets of data (AM 2019; AM 2020/2021; and SM4 2020/2021); we obtained the uncertainty associated with the different measures through a 10-fold cross-validation. This procedure was performed alongside the transfer learning for the two 2020/2021 test sets. We also report the variation in the metrics before and after data augmentation and/or transfer learning. Specifically, we measured the metric values of AM 2019 before and after data augmentation; for both AM and SM 2020/21 we measured the metric values before and after transfer learning with or without data augmentation. TL: transfer learning; DA: data augmentation.

	AM 2019 Test Set		AM 2020/2021 Test Set			SM4 2020/2021 Test Set		
TL	−		−	+		−	+	
DA	−	+	−	−	+	−	−	+
Accuracy	0.94 ± 0.01	0.90 ± 0.01	0.89 ± 0.01	0.96 ± 0.01	0.92 ± 0.02	0.92 ± 0.01	0.95 ± 0.01	0.91 ± 0.01
Precision	0.85 ± 0.04	0.59 ± 0.03	0.44 ± 0.01	0.81 ± 0.04	0.56 ± 0.04	0.59 ± 0.01	0.83 ± 0.02	0.52 ± 0.04
Recall	0.69 ± 0.02	0.82 ± 0.04	0.89 ± 0.02	0.71 ± 0.03	0.86 ± 0.02	0.75 ± 0.01	0.63 ± 0.03	0.84 ± 0.02
F1 Score	0.76 ± 0.02	0.69 ± 0.01	0.59 ± 0.01	0.76 ± 0.01	0.68 ± 0.02	0.66 ± 0.01	0.71 ± 0.01	0.64 ± 0.03

## Data Availability

Exemplars can be found in the Appendix A. The data presented in this study are available on request from the corresponding author.

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
