# Peer review of "There You Are! Automated Detection of Indris’ Songs on Features Extracted from Passive Acoustic Recordings"

_animals, 2023, doi:10.3390/ani13020241_

Round 1

Reviewer 1 Report

Overall:

This manuscript explores methodology to allow for the evaluation of bioacoustics data in a more efficient and economical way.  The introduction does a good job of providing the context for how this technology is being used in other species and providing a justification for its potential utility in Indri.  The methodology employed seemed effective at creating a program with high accuracy and recall. The manuscript then employed this program to evaluate temporal patterns in the call based on time of day and season. The discussion did a good job of highlighting the limitations of this work but also identifying areas of potential application and further research.

 Simple Summary:

Line 22: contribute to which research field? Suggest clarifying

Line 22: Tense agreement- used present tense here “contributes” but then past tense “was” later in the sentence

Abstract:

Line 32: Suggest choosing a different word than “peculiar”- something like “characteristic or unique maybe?

Line 37: should Accuracy and Recall be capitalized?

Line 39: In the abstract it says that this methodology was used to evaluate spatial distribution of this species. You discuss the pattern of singing behavior in relation to time of day and season, however you do not really address how this model was used to evaluate spatial distribution of Indri other than in Figure 2. Since this is something you discuss in the abstract it should be further elaborated on in the material and methods, results and discussion, or that particular aspect of the way this methodology is used should be removed from the abstract.  

Graphical abstract:

Line 56: If this is meant to stand alone than CNN should be fully written out as Convolutional Neural Network before it is abbreviated

Introduction:

Line 66-68: sentence is worded a little awkwardly- specifically the “allows gathering” portion

Line 69: suggest remove the word “indeed”

Line 76: suggest replace “growingly” with “increasingly”

Line 77: you put an abbreviation for machine learning in parenthesis (ML), however you do not use this abbreviation anywhere else in the manuscript but do write out machine learning. I think the abbreviation is unnecessary

Line 77: be consistent with capitalization, other places in the manuscript machine learning is capitalized (lines 115 and 156). If it is not a proper noun or a specific program/product then it shouldn’t be capitalized.

Line 89-91: need a citation for this statement

Lines 95-97: need a citation for this statement

Line 103-105: suggest rewording a bit, potentially putting the examples in parenthesis… (e.g. data type, efficacy of data processing etc)

Line 108-109: need citation for this statement

Line 115: are machine learning and deep machine learning the same thing? If so then please be consistent in how you refer to this in other areas of the manuscript

Line 119-120: need citation for this statement

Line 120- 122: need citation for this statement

Line 136: be consistent with uses of abbreviation- sometimes you use TL and other times you spell out transfer learning again – should choose one or the other. Similar comment CNN- sometimes you spell it out and other times you abbreviate it

Line 138-142: unclear if this study utilized the Hainan gibbon call for TL to try and classify black and white ruffed lemur calls- can you expand and clarify this a bit? Also need citations for this statement

Line 146: “tremendous” is not the best word choice, suggest using a different word

Line 148: the placement of this species name and citation is a little unusual- if this is an example of how this technology is already being employed in lemur species that needs to be explained a bit better

Line 156-158: sentence is a little stilted- suggest rewording

Line 158: suggest adding the word “species” after lemur

Line 158-160: this wording is a little confusing- if we don’t know the population size how do we know that populations are decreasing? Suggest rewording

Line 160: suggest removing the word “indeed” and just start the sentence with “This species…

Line 167-170: suggest rewording, this sentence is very long and a bit unclear and it is a really crucial piece of information because it sets up the reason why you are using this species

Line 173-178: Clarify a bit- reading this it sounds like you had two objectives- make them clearer and more concise and suggest taking out words like “In particular” and “Lastly”

Material and Methods

Line 184: need to spell out abbreviations the first time they are used- since this is the only time this abbreviation is used it would be better to just write out “above sea level”

Line 185: again question whether you need to use an abbreviation if you never use it again the rest of the manuscript

Line 188: does the publication require product information? If so please include info for the recorders

Line 188: suggest removing the word Specifically and just starting the sentence “In 2019…”

Line 201: suggest removing the word “indeed”

Line 201-205: it seems like the intention was to have this be one sentence instead of 2 – suggest rewording since it doesn’t completely make sense as it is currently worded

Line 210: were the classifications for AM 2020/2021 and SM4 2020/2021 also verified by other researchers in the same way that AM 2019 was? If so that needs to be included

Line 210-212: this is a little unclear- so there were 250 selected samples taken from amongst the three data sets that both observers looked at for inter-observer variability?

Line 212: if this journal requires product information please include the product info for “Python”

Line 218: suggest rewording-the first part of the sentence is a little awkward

Line 247: this is the first time you define the abbreviation for data augmentation (DA) but you use the term earlier in the paper and also still spell it out later in the manuscript- if you feel like this abbreviation is needed I would suggest defining it earlier and then using it throughout the text- personally I think that excessive abbreviations can make manuscripts a bit more difficult to read so it may be worth really considering when abbreviations are needed and when the entire term can be spelled out.

Discussion:

Line 376- 380: suggest rewording this sentence for conciseness- it is a really important point but the sentence is quite long and a bit confusing as written

Line 402: suggest removing or rewording the end of the sentence that says “when affecting it”- doesn’t really make sense as written

Conclusions:

Line 462: suggest rewording for clarity

Line 479: this is the conclusion of your work, is a citation needed?

Table 1: It was a little unclear which data were associated with which conditions since some of the “minus” marks were not directly above a data set- maybe adding a vertical lines to better delineate which data fell into which category or putting designation of the TL status directly above each data column

Author Response

Overall:

This manuscript explores methodology to allow for the evaluation of bioacoustics data in a more efficient and economical way. The introduction does a good job of providing the context for how this technology is being used in other species and providing a justification for its potential utility in Indri.  The methodology employed seemed effective at creating a program with high accuracy and recall. The manuscript then employed this program to evaluate temporal patterns in the call based on time of day and season. The discussion did a good job of highlighting the limitations of this work but also identifying areas of potential application and further research.

We thank the reviewer for this positive assessment of our work!

Simple Summary:

Line 22: contribute to which research field? Suggest clarifying
We clarified this: ‘Our study contributes significantly the automated wildlife detection research field’. New L22

Line 22: Tense agreement- used present tense here “contributes” but then past tense “was” later in the sentence
We modified ‘it was a first attempt’ as ‘it represents a first attempt’. New L23

Abstract: 

Line 32: Suggest choosing a different word than “peculiar”- something like “characteristic or unique maybe?
We followed the suggestion and replaced the word ‘peculiar’ with ‘distinctive’. The sentence now reads: “We first used indris’ songs, loud distinctive vocal sequences, to detect the species’ presence”. New LL 30-31

Line 37: should Accuracy and Recall be capitalized?
We used the lowercase letter throughout the text. See new L36 for instance

Line 39: In the abstract it says that this methodology was used to evaluate spatial distribution of this species. You discuss the pattern of singing behavior in relation to time of day and season, however you do not really address how this model was used to evaluate spatial distribution of Indri other than in Figure 2. Since this is something you discuss in the abstract it should be further elaborated on in the material and methods, results and discussion, or that particular aspect of the way this methodology is used should be removed from the abstract.  
We agree with the reviewer and rephrased the sentence, which now reads: “For the three datasets, our network detected the song presence with high accuracy (>90%) and recall (>80%) values, and provided data about both daily and annual habits of indris' singing pattern, critical information to optimise the field data collection”. New LL35-39

Graphical abstract:

Line 56: If this is meant to stand alone than CNN should be fully written out as Convolutional Neural Network before it is abbreviated
We agreed with the suggestion and modified the sentence accordingly: “During this process, we balanced instances of song presence/absence within the training set via data augmentation (DA); d) the 2019 dataset is employed to obtain a trained Convolutional Neural Network capable of performing automated detection”. New LL54-57

Introduction:

Line 66-68: sentence is worded a little awkwardly- specifically the “allows gathering” portion
We rephrased the sentence, which now reads: ‘Passive acoustic monitoring (PAM) enables the collection of information about species’ presence, abundance, density, distribution, population status, and dynamics; moreover, PAM allows for investigating seasonal and geographical variability of vocal behavior’. New LL66-69

Line 69: suggest remove the word “indeed”
Removed. The sentence now reads: “Using passive acoustics has undeniable advantages: it represents a non-invasive and relatively cheap technology in comparison to direct data collection, recorders are easy to deploy, and can be left in situ for extended periods”. New LL69-71

Line 76: suggest replace “growingly” with “increasingly”
Replaced as suggested. The sentence now reads: “Although most data are still manually processed, in recent years the use of deep learning for automating -or at least facilitating- the labelling and analysis of bioacoustic data is becoming increasingly relevant”. New LL74-76

Line 77: you put an abbreviation for machine learning in parenthesis (ML), however you do not use this abbreviation anywhere else in the manuscript but do write out machine learning. I think the abbreviation is unnecessary.
Line 77: be consistent with capitalization, other places in the manuscript machine learning is capitalized (lines 115 and 156). If it is not a proper noun or a specific program/product then it shouldn’t be capitalized.
We agree with the reviewer and removed the abbreviation throughout the text. See for instance new L77, L116

Line 89-91: need a citation for this statement
We added the relative reference: ‘(Dominica and Eastern Tropical Pacific, with 23 and 43 coda types and an accuracy of 97.5% and 93.6, respectively [8])’ New LL90-92

Lines 95-97: need a citation for this statement
We added the relative reference: ‘The results indicated an accuracy above 95.5% and when generalising the network performance on data recorded at different locations, the authors found a precision above 89.2% [2]’ New LL96-98

Line 103-105: suggest rewording a bit, potentially putting the examples in parenthesis… (e.g. data type, efficacy of data processing etc)
We simplified the sentence which now reads: ‘When employing these algorithms, however, the effectiveness of the best practices (e.g. data type, data processing, data augmentation) is still debated’ New LL104-105

Line 108-109: need citation for this statement
We added the relative reference: ‘The network provided indications of species activity and abundance in multiple ecosystems [5]’ New LL109-110

Line 115: are machine learning and deep machine learning the same thing? If so then please be consistent in how you refer to this in other areas of the manuscript
We recognize this was confusing and used machine learning throughout the text. See for instance new L116, L155

Line 119-120: need citation for this statement
We added the relative reference: ‘The network was not only able to detect calls -and noise- but also to classify call types and identify the callers [13]’ New LL119-120

Line 120- 122: need citation for this statement
We added the relative reference: ‘Interestingly, this study applied data augmentation methods (random noise and amplitude modification) as well, but the authors found that those did not improve the network performance [13]’ New LL 120-123

Line 136: be consistent with uses of abbreviation- sometimes you use TL and other times you spell out transfer learning again – should choose one or the other. Similar comment CNN- sometimes you spell it out and other times you abbreviate it
We chose to remove all the abbreviations to TL and DA throughout the text.

Line 138-142: unclear if this study utilized the Hainan gibbon call for TL to try and classify black and white ruffed lemur calls- can you expand and clarify this a bit? Also need citations for this statement.
Thank you for the suggestion, we rephrased the sentence as follows: “To address data scarcity issues related to CNN, often occurring while studying rare species, transfer learning, a technique using a model trained on a given dataset for predicting a second one, has been proposed. A recent study focused on targeting the calls of two critically endangered primate species, the Hainan gibbon (Nomascus hainanus) and the black-and-white ruffed lemur (Varecia variegata)[16]. It found that transfer learning starting from an all-purpose image database resulted in an improvement of the classification scores, a network easier to design, and a reduction of the manual annotation time for both species [16]” New LL135-139

Line 146: “tremendous” is not the best word choice, suggest using a different word
We replaced the term by using the word ‘dreadful’.  new L146

Line 148: the placement of this species name and citation is a little unusual- if this is an example of how this technology is already being employed in lemur species that needs to be explained a bit better
We recognize the sentence was unclear and removed the reference to the species. New LL147

Line 156-158: sentence is a little stilted- suggest rewording
Line 158: suggest adding the word “species” after lemur
Line 158-160: this wording is a little confusing- if we don’t know the population size how do we know that populations are decreasing? Suggest rewording
We agree and rephrased the sentence as follows: ‘Indri is a critically endangered lemur [18] and one of the world's 25 most endangered primates [21]. The population trend of the species is decreasing, mainly due to hunting and habitat destruction’. New LL156-159

Line 160: suggest removing the word “indeed” and just start the sentence with “This species…
Removed, see new L159

Line 167-170: suggest rewording, this sentence is very long and a bit unclear and it is a really crucial piece of information because it sets up the reason why you are using this species
We recognize the sentence was unclear and rephrased it as follows: ‘Hence, despite indris possess a rich vocal repertoire [24,25] we focused on their songs as target sounds. Songs are indeed peculiar and species-specific powerful vocal displays [26], used in the inter-group long distance communication [27] and travelling up to two km…’ New LL166-169

Line 173-178: Clarify a bit- reading this it sounds like you had two objectives- make them clearer and more concise and suggest taking out words like “In particular” and “Lastly”.
We agree and rephrased that portion as follows: ‘We first aimed to develop an easy-to-train, versatile and fast classifier able to target indris’ songs while identifying the best practices in the setting up of an efficient tool: which kind of data to feed to the network (i.e. waveform versus spectrogram images), the amount of data to be manually labelled, the effectiveness of employing data augmentation and transfer learning, for instance [4,16]. Our second aim was to use the developed tool to explore diel and seasonal variation in the calling pattern of the species.’ New LL171-176

Material and Methods

Line 184: need to spell out abbreviations the first time they are used- since this is the only time this abbreviation is used it would be better to just write out “above sea level”
We agreed and followed the suggestion. New LL194

Line 185: again question whether you need to use an abbreviation if you never use it again the rest of the manuscript
We removed the abbreviation; the sentence now reads: ‘…within the Ankeniheny-Zahamena Corridor, a large area of protected forest biome in eastern Madagascar.’ New LL194-195

Line 188: does the publication require product information? If so please include info for the recorders
We inserted the relative information as required by the journal. The sentence now reads: ‘we deployed ten AudioMoth recorders (LabMaker, Berlin, Germany), a low-cost, small-sized, and low-energy directional acoustic detector [32], and two Song Meter SM4 recorders (Wildlife Acoustics Inc., Maynard, MA, USA).’ New LL 197-199

Line 188: suggest removing the word Specifically and just starting the sentence “In 2019…”
Removed, see new L199

Line 201: suggest removing the word “indeed”
Removed, see new L212

Line 201-205: it seems like the intention was to have this be one sentence instead of 2 – suggest rewording since it doesn’t completely make sense as it is currently worded
We agree and rephrased the sentences as follows: ‘This frequency band has previously been used to study the indris’ song [36] since it contains its fundamental frequency (f0) and first harmonics. At the same time, this frequency bands minimizes the chosen range, allowing to mitigate the presence of other sounds which could be mistaken as a song’. New LL212-215

Line 210: were the classifications for AM 2020/2021 and SM4 2020/2021 also verified by other researchers in the same way that AM 2019 was? If so that needs to be included
See answer to next comment

Line 210-212: this is a little unclear- so there were 250 selected samples taken from amongst the three data sets that both observers looked at for inter-observer variability?
We recognize this was unclear and rephrased the sentence accordingly: ‘We visually inspected individually the spectrograms regarding the presence of the song: a single observer (DR) undertook the classification process for the AM 2019 dataset, then four researchers with field experience on indris’ activity verified this classification. To test for inter-observer reliability, two primary observers (DR and VF) evaluated 250 randomly selected files from the AM 2019 dataset, achieving a Cohen's kappa value of 0.9160 [37]. We computed Cohen's kappa via Python's module sklearn.metrics.cohen_kappa_score. Lastly, a different single observer with field experience (VF) undertook the classification process for the AM 2020/2021 and SM4 2020/2021 datasets’. New LL216-224

Line 212: if this journal requires product information please include the product info for “Python”
We included the product information (Python 2009; version 3; CreateSpace, Scotts Valley, CA, USA). Nel LL222

Line 218: suggest rewording-the first part of the sentence is a little awkward
We rephrased the sentence as follows: “We labelled the 13.7% of the AM 2019 dataset, the 9.3% of the AM 2020/2021 dataset, and the 9.7% of the SM4 2020/2021 dataset as files containing songs.” New LL230-232

Line 247: this is the first time you define the abbreviation for data augmentation (DA) but you use the term earlier in the paper and also still spell it out later in the manuscript- if you feel like this abbreviation is needed I would suggest defining it earlier and then using it throughout the text- personally I think that excessive abbreviations can make manuscripts a bit more difficult to read so it may be worth really considering when abbreviations are needed and when the entire term can be spelled out.
We agree and we chose to remove all the abbreviation to TL and DA throughout the text.

Discussion:

Line 376- 380: suggest rewording this sentence for conciseness- it is a really important point but the sentence is quite long and a bit confusing as written
We rephrased the sentence, which now reads: “At heart: in a successive manual classification of the positively classified cases, we will encounter a higher number of false positive occurrences. While we will easily discard them in a second manual classification, we are concurrently narrowing the false negatives, minimising the number of overlooked recordings containing songs” New LL408-412

Line 402: suggest removing or rewording the end of the sentence that says “when affecting it”- doesn’t really make sense as written
We rephrased the sentence as follows: ‘However, they also found that treating class imbalance through resampling either worsened the network performance, or did not affect it’. New LL429-431

Conclusions:

Line 462: suggest rewording for clarity
We agree and rephrased the sentence, which now reads: ‘Long-term monitoring generates enormous amount of data: the manual labelling of hundreds of hours of recordings is virtually unfeasible’. New LL50-501

Line 479: this is the conclusion of your work, is a citation needed?
We removed the reference. New L517

Table 1: It was a little unclear which data were associated with which conditions since some of the “minus” marks were not directly above a data set- maybe adding a vertical lines to better delineate which data fell into which category or putting designation of the TL status directly above each data column
We agree and edited the table according to the suggestion.

Reviewer 2 Report

This manuscript describes the mathematical algorithm for automated detection of songs of Madagascar primate Indri indri, developed on the basis of recordings obtained in the field with devices for passive acoustic recording throughout two years. The authors also claim presenting dynamics of diurnal and seasonal vocal activity of this species. Whereas the mathematical algorithm is described in detail, and its use is well-substantiated and discussed, the section describing the diurnal and seasonal vocal activity of indri is strongly subordinate, has not own methods and poorly introduced and discussed.

Title

Please add "indri" in the title. Primate is too uncertain.

Simple Summary

The title and simple summary do not provide to the reader an immediate knowledge of whether this paper is methodical on the software for automated data processing or provides scientific data on daily and spatial patterns of the vocal activity of indri. With two ideas mixed in one paper, both are lost. Please make clear, what part is methodical and what is resultative, as the uncertainty potentially shortens the number of the readers of this paper.

L 100-101 Here, the network reached an accuracy of 99.7% in the binary classification of noise and bat calls and of 96% for the sole species classification [9]

In this sentence, the authors indicated how the accuracy was estimated. In many other cases within Introduction, they just say “with accuracy of…” without indicating how the accuracy was estimated, relative background noises, other noises or calls of other species. Please indicate this throughout the Introduction, to make it clear for the reader.

L 203 delete “while”

Add values on frequency axis.

Author Response

This manuscript describes the mathematical algorithm for automated detection of songs of Madagascar primate Indri indri, developed on the basis of recordings obtained in the field with devices for passive acoustic recording throughout two years. The authors also claim presenting dynamics of diurnal and seasonal vocal activity of this species. Whereas the mathematical algorithm is described in detail, and its use is well-substantiated and discussed, the section describing the diurnal and seasonal vocal activity of indri is strongly subordinate, has not own methods and poorly introduced and discussed.

We agree with the reviewer and added the missing sections:
Introduction (new LL176-188)
Indeed, although indris are known to sing more during the breeding season [51] and certain part of the day [16] an accurate quantification of seasonal and diel trends of their singing activity is missing. Passive sensors can be used to describe vocal activity over time spans ranging from hours to seasons (Melo et al. 2021) and passive acoustics has indeed already been successfully employed for the description of circadian and seasonal trends of vocal activity of primate species. For instance, PAM has been used to gain ecological insights and to identify periods with the highest calling pattern, at diel and seasonal level in black and gold howler monkey (Alouatta caraya; Pérez‐Granados and Schuchmann 2020) and Guianan red howler monkey (Alouatta macconnelli, Do Nascimento et al. 2021). Moreover, providing precise information on the vocal behavior of the species is fundamental for designing appropriate monitoring programs. These can, in fact, ease data collection, reduce the sampling effort and therefore the amount of unnecessary data to storage and process (Melo et al. 2021).

Methods (new LL308-324)
Lastly, we employed the network to determine the calling pattern of the species in terms of both diel and seasonal activity,
deriving their respective probability distributions. Regarding the diel distribution, we used 48 30-minute bins, thus equalling the time interval between the start of two recordings. Although indri activity is tied to the sunrise [28], we decided to show the daily trend accumulated for the entire year instead of dividing it by the different seasons; at such latitude, in fact, the sunrise time range remains roughly constant [49]. Regarding the seasonal distribution, we converted the date associated with each recording to the respective week of the year, binning the year into 52 weeks. If a week contained less than 250 recordings, we considered it "unsampled" and therefore kept out its recordings from the computation of the probability of the sampled weeks. With this process, we obtained both the diel and the seasonal distributions from the 10 test sets of the AM 2020/2021 and SM4 2020/2021 datasets obtained from cross-validation. Then, separately for the two datasets, we used the ten values obtained for each bin to compute their respective means and standard deviations. Finally, we determined the expected diel and the seasonal distributions for the two different datasets via the manual classification performed on the overall dataset.

Discussion (new LL474-483)
From a future perspective, narrowing down where a song is given could allow a finely-tuned recognition of a particular group's position and identity. We were working with a limited array of sensors, but denser arrays can indeed inform about the precise placement of a group of singers (Blumstein et al. 2011). Instead, investigating the temporal patterning of songs can represent an indicator of the effect of perturbations over an indri population. We know that anthropogenic noise can affect the temporal occurrence of birdsong (Dominoni et al. 2016; Sierro et al. 2017) and singing patterns can provide researchers with essential information on the impact of the human alteration of natural habitats (Lajolo et al. 2008), a problem dramatically present in the current situation in Madagascar.

Title

Please add "indri" in the title. Primate is too uncertain.
We followed the suggestion. The current title is “There you are! Automated song detection on passive acoustic sensors provides insights into indris' vocal activity”

 Simple Summary

The title and simple summary do not provide to the reader an immediate knowledge of whether this paper is methodical on the software for automated data processing or provides scientific data on daily and spatial patterns of the vocal activity of indri. With two ideas mixed in one paper, both are lost. Please make clear, what part is methodical and what is resultative, as the uncertainty potentially shortens the number of the readers of this paper.

L 100-101 Here, the network reached an accuracy of 99.7% in the binary classification of noise and bat calls and of 96% for the sole species classification [9]. In this sentence, the authors indicated how the accuracy was estimated. In many other cases within Introduction, they just say “with accuracy of…” without indicating how the accuracy was estimated, relative background noises, other noises or calls of other species. Please indicate this throughout the Introduction, to make it clear for the reader.
We followed the suggestion and added the missing information:
New LL 83-85 “For instance, CNN have been successfully trained to classify among presence and absence of different kinds of sounds, taxonomic groups, and environments…”

New LL89-94 “Indeed, the network has been trained to classify coda types recorded in different zones (Dominica and Eastern Tropical Pacific, with 23 and 43 coda types and the coda type classification achieved an accuracy of 97.5% and 93.6, respectively [8]). Moreover, in a further task of vocal clan discrimination, the network achieved an accuracy of 95.3% for two Dominican clans and 93.1% for four Eastern Tropical Pacific clans [8].

New LL 96-97 “The results indicated an accuracy above 95.5% in classifying between noise and one vocal type”

L 203 delete “while”
Deleted. New L214

Sample_spectrogram.pdf: add values on frequency axis.
Added

Reviewer 3 Report

In general, I must say that the paper focus on an interesting and important topic, automated song or call recognition. The problem is to judge the specific advantages of their automated approach.

1) The authors stated in abstract that it is possible to get circadian, circannual distribution of indri songs as well as spatial distribution with the help of their CNN approach.

Every automated recognition software can give circadian, circannual distribution if they get the appropriate data. I also did not find anything in the model which could be able to make some predictions of the spatial distribution. Maybe the author can point out clearly which information are generated by the model and which were given to the model.

2) The authors could demonstrate convincingly that their CNN approach is an effective tool to detect indri songs. Unclear are further advantages. If I understand correct, one advantage is data reduction which helps to improve model performance.

This should help also other researchers studying other species (abstract & conclusion). But it remains unclear to which extend the data reduction works with other song or call structures.

It would be helpful to get some information on this point.

3) Maybe one reason that I have problems to valuate this approaches is that there is no clear comparison between the results found by the human observers and the CNN results. Fig. 2 & 3 are not really helpful.

4) It would be also nice to have more information about the quality of the original recordings (noise, reverberation, other animal sounds). The distance between recording units and source. And how lower sound quality, higher distances influence the possibility to make such a strong data reduction and get reliable results out of it.

Minor points:

Title “There are you! ….”:

1) This is a bit irritating. If I understand it correct, the CNN itself just can count the number of song and not the amount of callers or the location of the songs.

2) Intro “In particular, our first aim was to develop an easy-to-train, versatile and fast classifier able to 173 target indris’ songs while identifying the best practices in the setting up of an efficient tool: which 174 kind of data to feed to the network (i.e. waveform versus spectrogram images), …”

Is it really very fast starting with PRAAT to make spectrograms? I know classifier which classify calls directly.

Author Response

In general, I must say that the paper focus on an interesting and important topic, automated song or call recognition. The problem is to judge the specific advantages of their automated approach. 

1) The authors stated in abstract that it is possible to get circadian, circannual distribution of indri songs as well as spatial distribution with the help of their CNN approach.

Every automated recognition software can give circadian, circannual distribution if they get the appropriate data. I also did not find anything in the model which could be able to make some predictions of the spatial distribution. Maybe the author can point out clearly which information are generated by the model and which were given to the model.
Also to answer to comments from reviewer #2 we modified the abstract and expanded the methods section to include more precise information regarding data which are given to the model (see new LL 308-324). We also removed from the abstract the reference to the spatial distribution. Moreover, we partially disagree with the reviewer. We do believe that ‘Every automated recognition software can give circadian, circannual distribution if they get the appropriate data’ (if they work). Instead, thanks not only to our performance metrics values, but also to the fact that the forecast singing pattern followed the actual trends, we can corroborate that our method is robust and actually able to depict biologically relevant information. Regarding the ‘predictions on the spatial distribution’, the amount of songs is directly related to the amount of indri groups (Pollock 1975). This kind of information can certainly be used to gain insights on the behavioural ecology of the species, for instance in terms of occupancy of given forest areas. However, we clearly stated that we did not directly test potential differences in song detection among recording locations (Discussion, LL484-487) and our is a discussion of our results which could open new research perspectives.

2) The authors could demonstrate convincingly that their CNN approach is an effective tool to detect indri songs. Unclear are further advantages. If I understand correct, one advantage is data reduction which helps to improve model performance.
Maybe this was unclear but the employment of the CNN is not directly related to the choice of applying feature extraction (referred to as data reduction). As explained on new LL245-247 and new LL420-423 the feature extraction is instead critical for fastening the data processing and granting the possibility for data exchange among research centres.

This should help also other researchers studying other species (abstract & conclusion). But it remains unclear to which extend the data reduction works with other song or call structures. It would be helpful to get some information on this point.
In the conclusions (new LL505-507 and new LL515-517) we tried to clarify that such a method should be employable on species owning similar vocal patterns, such as loud calls, for which we believe feature extraction could prove effective. Indeed, such displays are usually long vocal sequences, for which processing the whole data (e.g. spectrograms, waveforms...) would be extremely time-consuming.

3) Maybe one reason that I have problems to valuate this approaches is that there is no clear comparison between the results found by the human observers and the CNN results. Fig. 2 & 3 are not really helpful.
We are not sure we understood the reviewer’s point. However, our aim was not to compare the efficacy of our (human) classification with that of our tool. We aimed instead at building a pre-processing tool which would be able to reduce the time spent in manual classification, as stated throughout the manuscript (see for instance LL39-41 and LL408-412).

4) It would be also nice to have more information about the quality of the original recordings (noise, reverberation, other animal sounds). The distance between recording units and source. And how lower sound quality, higher distances influence the possibility to make such a strong data reduction and get reliable results out of it.
The reviewer is right: of course recordings of a pristine rainforest show a certain degree of background noise (for instance rain, presence of other species) which we are in the process of measuring but, unfortunately, we are not able to provide yet. Moreover, being the recordings made through passive acoustics, we cannot know the actual distance between recording units and source. We do know that indris’ songs can travel up to two km throughout the forest (Pollock 1975), but to have a correct assessment we would need the dbSPL of songs emission at the source, which we are not able to give nor to estimate. Furthermore, we did not estimate the ability of our network of detecting songs emitted far away from the recording units, which are very feeble on the spectrum. This is surely an aspect deserving to be investigated but that falls outside the aim of this work.

Minor points:

Title “There are you! ….”:

1) This is a bit irritating. If I understand it correct, the CNN itself just can count the number of song and not the amount of callers or the location of the songs.

We would like to clarify that ‘there you are’ refers to the capability of song detection among hundreds of hours of recording. In particular, the network identifies the presence or absence of songs, not their amount. Indris’ songs are a direct index of the species presence and songs features are directly related with the number of callers (Torti et al 2018, PlosOne). We can, therefore, reliably state that numbers of songs and number of callers are directly related with one another. Nonetheless, identifying the number of callers was not one of our aims and we do not think we refer to it nowhere in the manuscript, nor we believe the sentence the reviewer is referring to (‘there you are!’) does. We would like, therefore, to stick with this part proposed title, while we did already change the other part.

2) Intro “In particular, our first aim was to develop an easy-to-train, versatile and fast classifier able to 173 target indris’ songs while identifying the best practices in the setting up of an efficient tool: which 174 kind of data to feed to the network (i.e. waveform versus spectrogram images), …”

Is it really very fast starting with PRAAT to make spectrograms? I know classifier which classify calls directly.

We agree that PRAAT might not be the fastest alternative possible. Nonetheless, our results clearly show that after the initial manual classification, the data necessary for further analyses are extremely reduced. By consequence, if we were to process data collected after 2021 (which is exactly our case) once the network has been perfected, we would need to process just a small part of the new data. We are aware of the robustness of our working flow, but we are open to test direct classifiers in future work.

Round 2

Reviewer 3 Report

no further comments